# Investigation of the impact of commonly used medications on the oral microbiome of individuals living without major chronic conditions

**Vanessa DeClercq**[1,2]*, **Jacob T. Nearing**[3], **Morgan G. I. Langille**[1,3]

**1** Department of Pharmacology, Dalhousie University, Halifax, Nova Scotia, Canada, **2** Department of Community Health and Epidemiology, Dalhousie University, Halifax, Nova Scotia, Canada, **3** Department of Microbiology and Immunology, Dalhousie University, Halifax, Nova Scotia, Canada

* vanessa.de.clercq@dal.ca

**Data Availability Statement:** The sequencing data analyzed during the current study have been uploaded to the European Nucleotide Archive and are available under the accession number

## Abstract

### Background

Commonly used medications produce changes in the gut microbiota, however, the impact of these medications on the composition of the oral microbiota is understudied.

### Methods

Saliva samples were obtained from 846 females and 368 males aged 35–69 years from a Canadian population cohort, the Atlantic Partnership for Tomorrow's Health (PATH). Samples were analyzed by 16S rRNA gene sequencing and differences in microbial community compositions between nonusers, single-, and multi-drug users as well as the 3 most commonly used medications (thyroid hormones, statins, and proton pump inhibitors (PPI)) were examined.

### Results

Twenty-six percent of participants were taking 1 medication and 21% were reported taking 2 or more medications. Alpha diversity indices of Shannon diversity, Evenness, Richness, and Faith's phylogenetic diversity were similar among groups, likewise beta diversity as measured by Bray-Curtis dissimilarity ($R^2$ = 0.0029, $P$ = 0.053) and weighted UniFrac distances ($R^2$ = 0.0028, $P$ = 0.161) were non-significant although close to our alpha value threshold ($P$ = 0.05). After controlling for covariates (sex, age, BMI), six genera (Saprospiraceae uncultured, *Bacillus*, *Johnsonella*, *Actinobacillus*, *Stenotrophomonas*, and *Mycoplasma*) were significantly different from non-medication users. Thyroid hormones, HMG-CoA reductase inhibitors (statins) and PPI were the most reported medications. Shannon diversity differed significantly among those taking no medication and those taking only thyroid hormones, however, there were no significant difference in other measures of alpha- or beta diversity with single thyroid hormone, statin, or PPI use. Compared to participants taking no medications, the relative abundance of eight genera differed significantly in

PRJEB38175. Code used to analyze all data is available at https://github.com/vdeclercq/ DECLERCQ_et_al_2021_Oral_Microbiome_ Medication..git Metadata used in this project cannot be shared publicly because participant consent and ethical restrictions do not permit public sharing of the data. Metadata are available from Atlantic PATH Data & Biological Samples Access Committee (contact via Ellen. Sweeney@dal.ca) for researchers who meet the criteria for access to confidential data and have received approval from a Research Ethics Board (REB). Additional information can be obtained by contacting info@atlanticpath.ca.

**Funding:** The author(s) received no specific funding for this work.

**Competing interests:** The authors have declared that no competing interests exist.

participants taking thyroid hormones, six genera differed in participants taking statins, and no significant differences were observed with participants taking PPI.

## Conclusion

The results from this study show negligible effect of commonly used medications on microbial diversity and small differences in the relative abundance of specific taxa, suggesting a minimal influence of commonly used medication on the salivary microbiome of individuals living without major chronic conditions.

## Introduction

Knowledge of how microbes contribute to human health and disease is increasing rapidly and over the last several years we have gained insight into how microbes interact with certain medications. Medications may alter the composition of the microbiota, altering efficiency and producing side effects [1–3]. Several studies have demonstrated that commonly prescribed medications can alter the diversity/composition of the gut microbiota [4–8]. Furthermore, it has been shown that members of the gut microbiome can enzymatically alter various medications altering their efficacy [3]. Much of the current research on medication use and the gut microbiome focus on a hand full of medications such as antibiotics, diabetes medication, and protein pump inhibitors (PPIs) [4–8]. However, more recent research across multiple cohorts suggests that in addition to antibiotics, PPIs and metformin, multiple commonly prescribed medications such as anticholinergic inhalers, paracetabol, selective serotonin reuptake inhibitors (SSRI), laxatives, and opioids are associated with the gut microbiota composition and/or changes in specific taxa [9, 10].

Interestingly, in addition to PPIs altering the microbial composition of the gut, they also increase the relative abundance of oral bacterial species in the gut microbiome [5]. Importantly, the gut is not the only human microbiome influenced by medication use. PPI treatment may influence microbial communities within different areas of the body such as the oral cavity [11, 12], esophagus [11], and stomach [13]. For example, PPI treatment has been reported to alter bacterial composition by reducing alpha diversity and altering the abundance of specific taxa in the oral cavity [12].

The oral cavity represents the initial part of the digestive tract and has a unique and diverse microbial composition that is reflective of distinct niches such as the teeth, cheek, hard palate, tongue and saliva [14–16]. The saliva includes microbiota dethatched from the various niches of the oral cavity and exhibits overlapping microbial profiles with several oral niches [16, 17]. The oral microbiome plays an important role in maintaining oral health homeostasis by supporting oral health or contributing to local conditions such as periodontitis, dental caries, and endodontic infections [18, 19] as well as diseases such as diabetes, obesity, cancer, and inflammatory bowel disease [14, 18, 20].

Our previous research demonstrates the composition of the salivary microbiome is associated with several sociodemographic, lifestyle, and anthropometric factors [21]. Importantly, this work established that the above factors were only able to explain a small extent of the variability between samples, indicating that each of these factors on their own contributes little to overall oral microbial diversity and that other factors must also contribute to the composition of an individual's oral microbiome. One factor contributing to differences in microbial composition may be prescription medications and although much of the previous research has

focused on the gut microbiome [2, 4, 6, 8–10], multiple medications have been reported to impact oral health causing symptoms such as dry mouth, lesions, ulcers, and altered taste [22, 23]. Therefore, it is likely that medication use may also be disrupting the oral microbial community. Research suggests that administration of PPI alters salivary microbiota diversity [12] and shifts the proportions of relative taxa [24]. Research on medication use and the oral microbiota is extremely limited but knowing that prescription medications cause side effects in the oral cavity and augment the microbiome at other body sites, it is plausible that commonly used medications could influence the oral microbiome. Since the oral microbiome plays an important role in health and disease, research on factors such as common medications that may alter the oral microbiota are needed as they could have unintended consequences on human health. Therefore, this study aimed to investigate the role of commonly used medications on the composition and diversity of the oral microbiota of adults taking single or multiple medications, as well as the most commonly reported medications in the cohort. We hypothesized that the use of medications would be associated with changes in oral microbiota composition and diversity, and further augmented with the use of multiple medications.

## Materials and methods

### Ethics

Ethics approval for the Atlantic Partnership for Tomorrow's Health (PATH) study was given by the appropriate provincial and regional ethics boards in each Atlantic province (New Brunswick: Horizon Health Network and Vitalité Health Network; Nova Scotia: Nova Scotia Health Authority Research Ethics Board and IWK Research Ethics Board; Newfoundland and Labrador: Health Research Ethics Board Newfoundland; Prince Edward Island: Health Prince Edward Island). All participants provided written informed consent before participation in the study. This research has been conducted using Atlantic PATH data and biosamples, under application #2018–103.

### Cohort description and study design

The use of a cross-sectional study design allowed us to conduct a rapid analysis and gain a glimpse into the role of the oral microbiome from a population-based cohort while comparing multiple drugs at the same time and controlling for other variables. This cross-sectional study examined participant data from the Atlantic PATH cohort. Atlantic PATH is part of the Canadian Partnership for Tomorrow's Health (CanPath) Project, a national prospective cohort study examining the influence of genetic, environmental, and lifestyle factors in the development of chronic disease. Between 2000–2019 CanPath recruited participants between the ages of 30–74 years. Further details on recruitment and data collection have been previously detailed [25, 26]. At baseline, participants completed a standardized set of questionnaires (available at: https://www.atlanticpath.ca/) and a subset of participants also had anthropometric measures, including BMI, and biological samples collected. Biological samples included blood, urine, toenails, and saliva samples.

Over 8,000 saliva samples were collected; 1,214 samples were included in this study based on the inclusion criteria: (i) oral microbiota data available, (ii) non-smoker, and (iii) no major chronic conditions. Major chronic conditions were self-reported data and included any of the following conditions: hypertension, myocardial infarction, stroke, asthma, chronic obstructive pulmonary disease, major depression, diabetes, inflammatory bowel disease, irritable bowel syndrome, chronic bronchitis, emphysema, liver cirrhosis, chronic hepatitis, dermatologic disease (psoriasis and eczema), multiple sclerosis, arthritis, lupus, osteoporosis, and cancer.

Sociodemographic characteristics of these participants have been summarized and previously published [21].

## Medication data

Self-reported prescription medication data were collected through the baseline questionnaire. Participants were asked if they were currently taking any medications prescribed by a doctor and dispensed by a pharmacist (yes/no/don't know) and if yes, asked to provide the name of the medication along with the *drug* identification number (DIN). This information was used to code medications according to the Anatomical Therapeutic Chemical (ATC) Classification System at level 4, representing the chemical, therapeutic, and pharmacological subgroup (e.g., proton pump inhibitors) [27]. Participants were then grouped as none, single, or multi medication users according to classification at the fourth level of the ATC code. Participants that were taking one unique medication at ATC code level 4 were classified as a 'single' medication user, those taking ≥2 unique medications at the ATC code level 4 were classified as a 'multi' medication user and those that completed the questionnaire without listing any medications were assumed to not be taking any medications and classified as 'none'. Subsequently, the frequency of each reported medication at ATC code level 4 was assessed to determine the most commonly reported medications. Medications that were reported more than 5 times are listed in Table 1, with thyroid hormone medications, proton pump inhibitors, and HMG CoA reductase inhibitors (statins) being the 3 most frequently reported. Participants that were taking the most frequently reported medications were further divided into those that were only taking the specific medication or those that were taking the specific medication plus other medications (eg. Thyroid, Thyroid+, Statin, Statin+, PPI and PPI+). These specific medication groups and the none, single, and multi-medication groups were used for further statistical analysis on oral microbial composition and diversity.

**16S rRNA gene sequencing and analysis.** Raw 16S rRNA gene sequencing data were processed as previously described [21]. Briefly primer trimmed paired end reads were stitched together using VSEARCH [28] and quality filtering using QIIME2 [29]. Reads were trimmed to 360 bp and the QIIME2 Deblur plugin [30] was used to produce amplicon sequence variants (ASVs). ASVs found in an abundance less than 0.1% of the mean sample depth across all samples were filtered out. Taxonomy was assigned using a naïve Bayesian QIIME2 classifier trained on the 99% Silva V132 database [31–33]. Diversity and dissimilarity measures for alpha and beta diversity were generated as previously described by rarifying samples to 5000 reads [21], and samples with under 5000 reads were removed resulting in 1,049 samples for the statistical analysis.

## Statistical analysis

Statistical analysis was conducted using R Version 4.0.2. Chi-square analyses were used to determine significant associations between sex and medication use, and if significant was followed by pairwise Fisher's exact test. Differences in continuous variables such as age and BMI were analyzed using the Kruskal-Wallis test, if necessary, followed by pairwise comparisons with Dunn's tests (adjusted using Bonferroni correction). An alpha value of 0.05 was chosen for determining significance. Categorical variables are presented as frequency (counts) and percentage (%), and continuous variables are presented as medians and interquartile ranges (IQR). Statistical analysis of filtered taxonomic data (n = 1,049), including relative abundance plots, alpha and beta diversity comparisons, and principal coordinate analysis (PCoA) plots were performed using the R packages vegan and ggplot2.

**Table 1. Most frequently reported medication according to ATC code level 4.**

| Level 1 ATC code | Level 1 Drug Class | Level 4 ATC code | Level 4 Drug Class | Route of administration | Total | Single | Multi |
|---|---|---|---|---|---|---|---|
| A | Alimentary tract and metabolism | | | | | | |
| | | **A02BC** | **Proton pump inhibitors** | oral, parenteral | **108** | **38** | **70** |
| | | A02BA | H2-receptor antagonists | oral, parenteral | 14 | 5 | 9 |
| | | - | Other+ | | 17 | 3 | 14 |
| B | Blood and blood forming organs | | | | | | |
| | | B01AC | Platelet aggregation inhibitors | oral, parenteral, inhal. solution | 32 | 5 | 27 |
| | | B01AA | Vitamin K antagonists | oral, parenteral | 5 | 0 | 5 |
| | | - | Other+ | | 7 | 3 | 4 |
| C | Cardiovascular system | | | | | | |
| | | **C10AA** | **HMG CoA reductase inhibitors** | oral | **111** | **36** | **75** |
| | | C07AB | Beta blocking agents, selective | oral, parenteral | 10 | 2 | 8 |
| | | C10AB | Fibrates | oral | 6 | 3 | 3 |
| | | C10AX | Other lipid modifying agents | oral, parenteral | 6 | 0 | 6 |
| | | C07AA | Beta blocking agents, non-selective | oral, parenteral | 5 | 1 | 4 |
| | | C03AA | Thiazides | oral | 5 | 0 | 5 |
| | | - | Other+ | | 31 | 6 | 25 |
| D | Dermatologicals | | | | | | |
| | | D07AC | Corticosteroids, potent (group III) | topical | 7 | 1 | 6 |
| | | - | Other+ | | 21 | 3 | 18 |
| G | Genito urinary system and sex hormones | | | | | | |
| | | G03CA | Natural & semisynthetic estrogens | oral, nasal, rectal, transdermal, vaginal | 40 | 7 | 33 |
| | | G03DA | Sex hormones and modulators | oral, parenteral, rectal, vaginal | 19 | 1 | 18 |
| | | G03AA | Progestogens and estrogens, fixed combinations | varied routes | 17 | 10 | 7 |
| | | G02BA | Intrauterine contraceptives. | vaginal | 7 | 4 | 3 |
| | | G04BD | Drugs for urinary frequency and incontinence | oral, parenteral, transdermal | 7 | 1 | 6 |
| | | G03AB | Progestogens and estrogens, sequential preparations | varied routes | 6 | 3 | 3 |
| | | G03BA | 3-oxoandrosten (4) derivatives | oral, parenteral, rectal, sublingual/ buccal/oromucosal, transdermal | 6 | 0 | 6 |
| | | G03FA | Progestogens and estrogens | varied routes | 5 | 2 | 3 |
| | | G04CA | Natural and semisynthetic estrogens | oral | 5 | 2 | 3 |
| | | G04CB | Testosterone-5-alpha reductase inhibitors | oral | 5 | 0 | 5 |
| | | - | Other+ | | 17 | 3 | 14 |
| H | Systemic hormonal preparations, excluding sex hormones and insulins | | | | | | |
| | | **H03AA** | **Thyroid hormones** | oral, parenteral | **131** | **60** | **71** |
| | | - | Other+ | | 6 | 3 | 3 |
| J | Antineoplastic and immunomodulating agents | | | | | | |
| | | J05AB | Nucleosides and nucleotides excl. reverse transcriptase inhibitors | oral, parenteral | 7 | 1 | 6 |
| | | - | Other+ | | 6 | 3 | 3 |
| L | Antineoplastic and immunomodulating agents | | Other+ | | 5 | 2 | 3 |

(*Continued*)

**Table 1.** (Continued)

| Level 1 ATC code | Level 1 Drug Class | Level 4 ATC code | Level 4 Drug Class | Route of administration | Total | Single | Multi |
|---|---|---|---|---|---|---|---|
| M | Musculo-skeletal system | | | | | | |
| | | M01AE | Propionic acid derivatives | oral, parenteral, rectal | 19 | 11 | 8 |
| | | M05BA | Bisphosphonates | oral, parenteral | 19 | 4 | 15 |
| | | M01AH | Coxibs | oral, parenteral | 10 | 4 | 6 |
| | | - | Other[+] | | 20 | 5 | 15 |
| N | Nervous system | | | | | | |
| | | N06AB | Antidepressants -selective serotonin reuptake inhibitors | oral, parenteral | 62 | 24 | 38 |
| | | N06AX | Other Antidepressants | oral | 34 | 8 | 26 |
| | | N06AA | Non-selective monoamine reuptake inhibitors | oral, parenteral | 18 | 8 | 10 |
| | | N02CC | Selective serotonin (5HT1) agonists | oral, nasal, parenteral, rectal | 15 | 2 | 13 |
| | | N05CF | Benzodiazepine related drugs | oral | 13 | 3 | 10 |
| | | N05BA | Benzodiazepine derivatives | oral, sublingual/buccal/oromucosal parenteral, rectal | 9 | 0 | 9 |
| | | N02AA | Natural opium alkaloids | oral, parenteral, rectal | 7 | 2 | 5 |
| | | N06BA | Centrally acting sympathomimetics. | oral, parenteral | 7 | 2 | 5 |
| | | N03AX | Other antiepileptics | oral, parenteral | 6 | 1 | 5 |
| | | N03AE | Benzodiazepine derivatives | oral, parenteral | 5 | 1 | 4 |
| | | - | Other[+] | | 33 | 7 | 26 |
| P | Antiparasitic products, insecticides and repellents | | | | | | |
| | | - | Other[+] | | 2 | 0 | 2 |
| R | Respiratory system | | | | | | |
| | | R01AD | Corticosteroids | nasal | 29 | 8 | 21 |
| | | R06AX | Other antihistamines for systemic use | oral, parenteral | 6 | 2 | 4 |
| | | - | Other[+] | | 19 | 5 | 41 |
| S | Sensory organs | | | | | | |
| | | S01ED | Beta blocking agents | topical | 8 | 2 | 6 |
| | | S01EE | Prostaglandin analogues | topical | 8 | 4 | 4 |
| | | | Other[+] | | 6 | 2 | 4 |
| U | Unknown | - | unknown | | 19 | 5 | 14 |

[+]All other medications in this category that had less than 5 counts total and were combined into 'Other'.

Bold text indicates the top 3 most frequently reported medications at ATC Level 4.

## Differential abundance analysis

As previously described, there are a myriad of different packages for identifying differential abundant microbes in 16S rRNA gene sequencing data. Based on previous research on these tools we have found that presenting the results from several different tools can increase the interpretability and reproducibility of findings [34]. As such, differential abundance analysis of bacterial genera was conducted on non-rarified counts using four different tools designed for differential abundance analysis: Corncob version 0.2.0 [35], ALDEx2 version 1.22.0 [36], MaAsLin2 version 1.4.0 [37], and ANCOM-II version 2.1 [38]. For differential abundance testing, a prevalence cut-off filter was set to remove genera found in fewer than 10% of samples.

Furthermore, all differential abundance testing was done while controlling for covariates (sex, age, and BMI).

Differential abundance testing between different medication groups using Corncob was conducted using the "differentialTest" function and plotted as log odds. During this analysis in addition to controlling for the covariates described above, we also controlled for differential variability in taxonomic relative abundances. ALDEx2 package testing was done using the "aldex.glm" and "aldex.clr" functions. Default parameters were used along with a total of 128 Monte Carlo samplings to ensure robust test statistics. ANCOM-II testing was done using the ANCOM-II package available at https://github.com/FrederickHuangLin/ANCOM. The main "ANCOM" function was run with default parameters with a q value of 0.1. A taxa detected at the 0.8 limit was considered to be significantly associated with the tested medication group of interest. The MaAsLin2 R package was run using the function "maaslin2" with arcsine square root transformation and default parameters. In each case when applicable resulting p values were corrected for multiple hypothesis testing using the Benjamini and Hochberg algorithm. An alpha value of q = 0.1 was chosen as statistically significance.

## Diversity analysis

Four different alpha diversity metrics were assessed, which included Shannon diversity, evenness, Faith's phylogenic diversity, and richness (number of ASVs). Alpha diversity of each group was compared by Kruskal-Wallis test and if necessary, followed by pairwise comparisons with Dunn's tests and Bonferroni correction. Two beta diversity metrics, Bray-Curtis dissimilarity and weighted UniFrac distance, were analyzed using permutational multivariate analysis of variance (PERMANOVA; adonis2 vegan function) with 10,000 permutations while adjusting for covariates (sex, age, BMI). When PERMANOVA results were significant ($P < 0.05$), pairwise comparisons were conducted, and Bonferroni correction was applied to correct for multiple comparisons between groups.

## Results

### Cohort and medication use

We analyzed saliva samples from 1,214 individuals (n = 846 females, n = 368 males) aged 35–69 years with a median age of 56 [50–62] years and a BMI of 27 [24–30] kg/m$^2$. Of these individuals, 644 (53%) reported taking no medications, 318 (26%) reported taking 1 medication (single user) and 252 (21%) reported taking 2 or more medications (multi-medication user). Multi-medication users were taking up to 10 unique medications at ATC code Level 4. Multi-medication users were older (58 (52–62) years) than single (56 [50–61] years, $P = 0.002$) and non-medication users (56 [49–62] years, $P < 0.001$, S1 Table). BMI was 27 [24–30], 27 [24–30], and 27 [25–30] kg/m2 in non-medication, single, and multi-medication users, respectively ($P = 0.026$). Across medication groups, a similar proportion of females (67, 72, and 73%) and males (33, 28, and 27%) were found to be non-, single, and multi-medication users, respectively ($P = 0.129$).

Participants self-reported the use of 144 unique prescription medications (ATC level 4). The most frequently reported medications are listed in Table 1. The top three most frequently reported medication classes (Level 4 ATC code) were thyroid hormones, HMG-CoA reductase inhibitors (statins) and proton pump inhibitors (PPI). One hundred and twenty-seven participants reported taking thyroid medications, and of those 91% were females and 9% were male (S2 Table). Compared to participants taking no medications (56 years [49–62]), participants taking thyroid medications plus other medications were older (59 years [54–62], $P = 0.018$), but similar to those taking only thyroid medication (55 years [50–59], $P = 1.000$). The median

BMI ($P<0.891$) of participants taking no medication, only thyroid medication, or thyroid plus other medications were (27kg/m$^2$ [24–30], 27 [23–30], 27 [24–30], respectively). One hundred and eleven participants reported taking statin medication, and of those 51% were females and 49% were male. Compared to participants taking no medications, participants taking only statin medication ($P = 0.026$), or statin plus other medications ($P<0.001$) were older (56 years [49–62], 59 years [56–63], 61 years [57–64], respectively). The median BMI ($P<0.076$) of participants taking no medication, only statins medication, or statin plus other medications were similar (27kg/m$^2$ [24–30], 28 [25–29], 28 [26–30], respectively). One hundred and six participants reported taking PPI, and of those 72% were females and 28% were male. Participants taking no medications, those only taking PPI or PPI plus other medications were similar ages age (56 [49–62], 54 [49–61], 58 [52–62] years, respectively, $P = 0.143$) but had statistically different BMI values (27 [24–30], 29 [26–32], 28 [26–30] kg/m$^2$, respectively, $P< 0.001$).

## Microbial composition and diversity of medication users

To investigate whether the oral microbiota was altered in individuals taking prescription medications, we analyzed alpha and beta diversities as well as the microbial relative abundances obtained from saliva sample 16S rRNA gene sequencing data of medication and non-users. The alpha diversity indices of Shannon diversity ($P = 0.129$), Faith's phylogenetic diversity ($P = 0.062$), richness ($P = 0.210$) and Evenness ($P = 0.185$) were not statistically different among groups (Fig 1). Although the association with beta diversity and general medication was close to our alpha value threshold ($P = 0.05$), it was not statistically significant as measured by Bray-Curtis dissimilarity ($R^2 = 0.0029$, $P = 0.051$) and weighted UniFrac distances ($R^2 = 0.0028$, $P = 0.164$) controlling for sex, age, and BMI (Fig 2).

Using Corncob, which models the relative abundance of taxa using beta-binomial models, the relative abundance of nine genera (*Bacteroides*, Saprospiraceae uncultured, Flavobacteriaceae unclassified, *Bacillus*, *Johnsonella*, Burkholderiaceae unclassified, *Actinobacillus*, *Stenotrophomonas*, and *Mycoplasma*) were significantly different from non-medication users in the unadjusted model. However, after controlling for covariates (sex, age, BMI) only six genera (Saprospiraceae uncultured, *Bacillus*, *Johnsonella*, *Actinobacillus*, *Stenotrophomonas*, and *Mycoplasma*) remained significantly different (Fig 3). The significant abundance coefficients from the differential test conducted with Corncob are reported in S3 Table. Additional differential abundance tests using ALDEx2, MaAsLin2, and ANCOM2 were also conducted. Of the above genera identified as differentially abundant by Corncob, only *Johnsonella and Actinobacillus* were identified as differentially abundant by MaAsLin2 and no genera were identified using ALDEx2 and ANCOM-II (S3 Table).

## Influence of specific medications on microbial diversity

Finally, we examined oral microbial diversity and composition in participants taking the most frequently reported medications (thyroid hormone/statin/PPI) either alone or in combination with other medications. First, diversity among participants only taking thyroid hormones, statins or PPIs was assessed. Some measures of alpha diversity differed among thyroid hormone users (Shannon diversity $P = 0.041$; Evenness $P = 0.013$; S1A and S1B Fig) but beta diversity was non-significant among participants taking only thyroid hormones, statins, PPIs or no medications (Bray-Curtis P = 0.104; weighted UniFrac p = 0.324; S1C and S1D Fig).

Next, we explored possible interactions with other medications by analyzing microbial diversity in participants taking thyroid hormone/statin/PPI in combination with other medications. The Kruskal-Wallis test revealed overall significance with thyroid hormones users and Shannon diversity ($P = 0.011$) and Evenness ($P = 0.013$). Subsequently, Dunn's pairwise

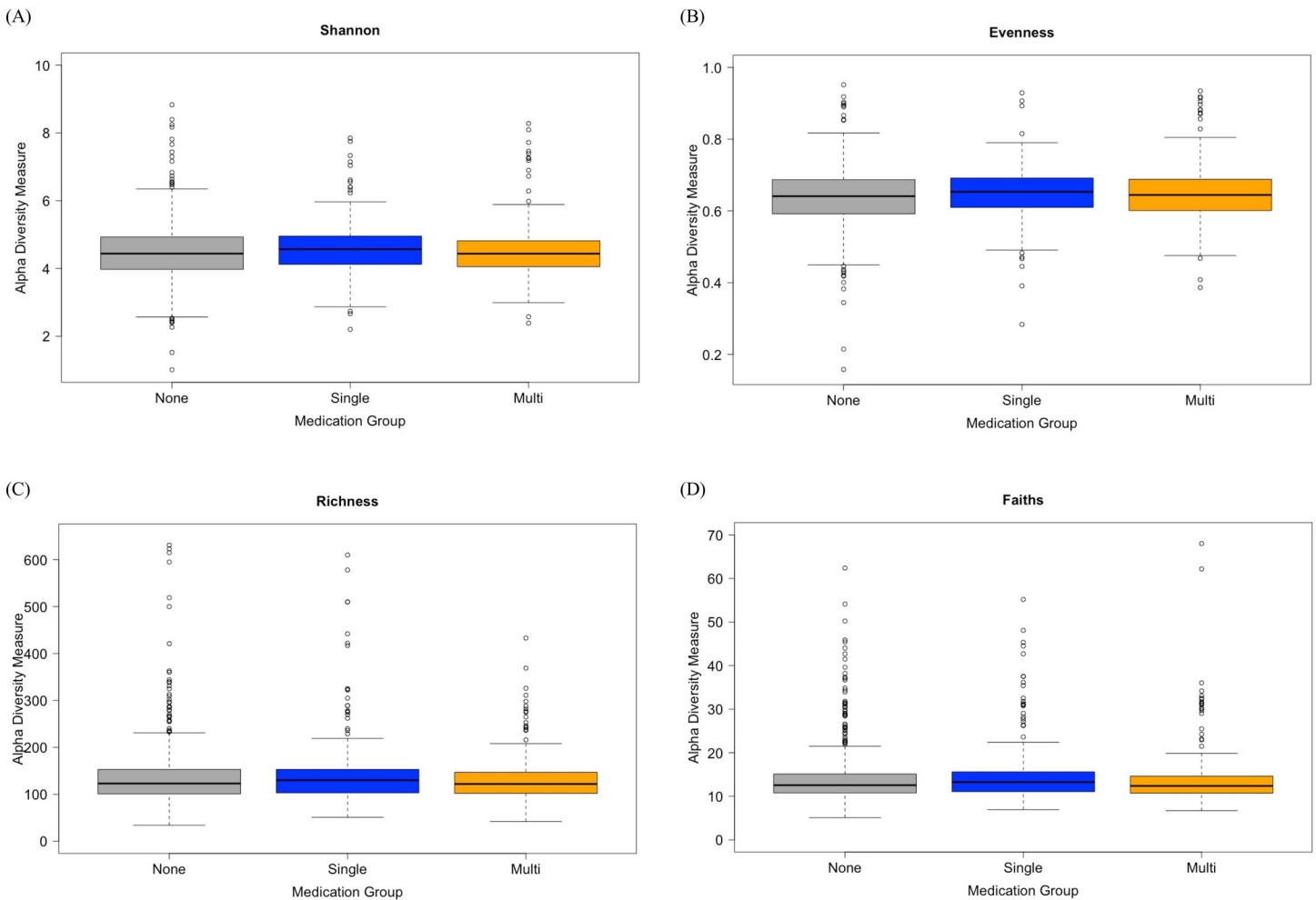

**Fig 1.** Four different alpha diversity metrics, (A) Shannon diversity, (B) Faith's phylogenetic diversity, (C) evenness, and (D) richness. No significant differences were found when using a Kruskal-Wallis test. "None" represents participants taking no medications (n = 546); "Single" represents participants taking only one medication at ATC code Level 4 (n = 274); "Multi" represents participants taking 2 or more medications at ATC code Level 4 (n = 225).

comparison showed that compared to those taking no medications, Shannon diversity ($P = 0.011$; Fig 4A) and Evenness ($P = 0.010$) differed significantly to those only taking thyroid hormones but was similar to those taking thyroid hormones in combination with other medication (Thyroid+). There was no significant difference in Shannon diversity and Evenness among Statin and Statin+ users ($P = 0.907$ and $P = 0.250$, respectively) or PPI and PPI+ users ($P = 0.950$ and $P = 0.798$, respectively) (Fig 4). The specific drug classes alone or in combination with other medications showed no differences in additional alpha diversity indices such as Faith's phylogenetic diversity (Thyroid $P = 0.483$; Statins $P = 0.270$; PPI $P = 0.582$), or richness (Thyroid $P = 0.156$; Statin $P = 0.572$; PPI $P = 0.709$). We did not find a significant association with thyroid hormone or statin medications, alone or in combination, with beta diversity as measured by Bray-Curtis dissimilarity (Thyroid Hormones $R^2 = 0.0041$ $P = 0.103$; Statins $R^2 = 0.0044$, $P = 0.077$; PPI $R^2 = 0.0048$, $P = 0.048$; Fig 5) and weighted UniFrac distances (Thyroid Hormones $R^2 = 0.0043$, $P = 0.166$; Statins $R^2 = 0.0057$, $P = 0.065$; PPI $R^2 = 0.0067$, $P = 0.038$; S2 Fig) after controlling for sex, age, and BMI. In contrast, there was a significant difference in Bray-Curtis dissimilarly ($R^2 = 0.0048$, $P = 0.048$; Fig 5) and weighted UniFrac

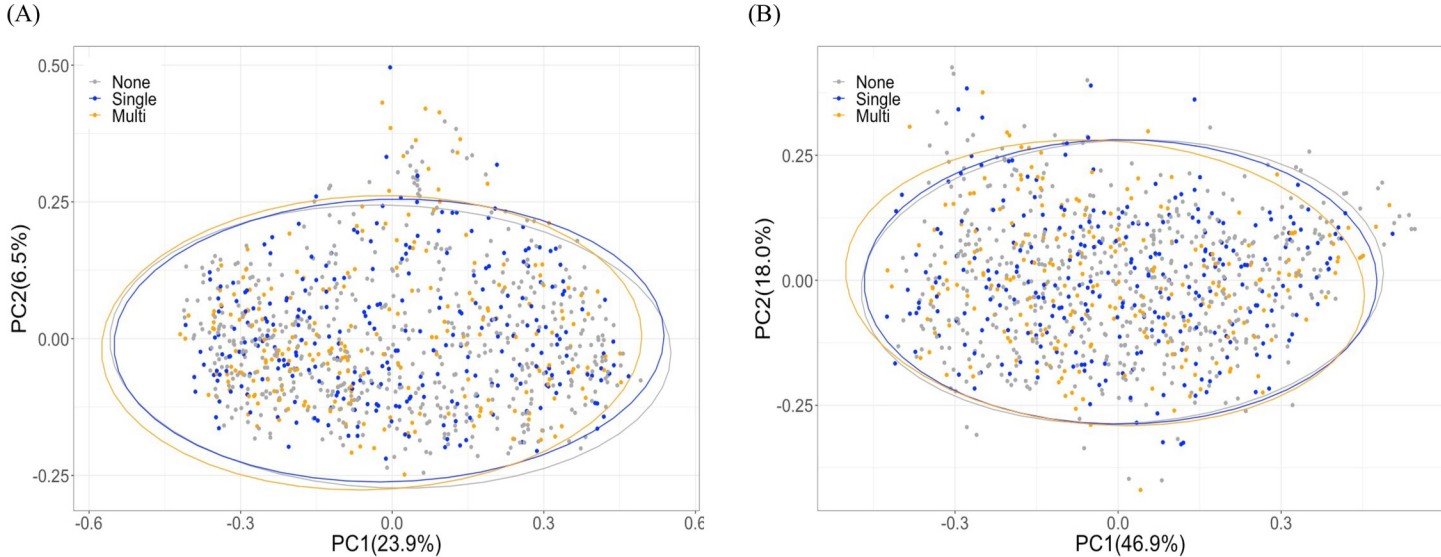

**Fig 2.** Beta diversity analyses among medication and non-medication users are represented by Principal Coordinates Analysis plots based on (A) Bray-Curtis dissimilarity and (B) weighted UniFrac. "None" (grey dots) represents participants taking no medications (n = 546); "Single" (blue dots) represents participants taking only one medication at ATC code Level 4 (n = 274); "Multi" (orange dots) represents participants taking 2 or more medications at ATC code Level 4 (n = 225).

distances ($R^2$ = 0.0067, $P$ = 0.038; S2 Fig) among PPI users after controlling for sex, age, and BMI. Subsequent pairwise comparison revealed this microbial diversity was driven by differences between non-medication users and PPI+ users (Bray-Curtis $P$ = 0.485 for None vs PPI, $P$ = 0.092 for None vs PPI+ and $P$ = 1.000 for PPI vs PPI+; weighted UniFrac $P$ = 0.949 for None vs PPI, $P$ = 0.040 for None vs PPI+ and $P$ = 0.912 for PPI vs PPI+).

At the genus level, the relative abundance of several genera between participants taking no medications compared to those taking thyroid hormones were identified as being statistically different using the R package Corncob (*Bacteroides*, *Prevotella 6*, *Tannerella*, Saprospiraceae uncultured, *Bergeyella*, *Bacillus*, Veillonellaceae uncultured, and *Mycoplasma*; S3A Fig). Using this approach Statin use was also found to be associated with the relative abundance of several taxa (*Bacteroides*, *Bacillus*, *Catonella*, *Johnsonella*, *Neisseria*, *Stenotrophomonas*; S3B Fig), There were no statistical differences in those taking PPI (FDR q>0.1) after controlling for the covariates sex, age, and BMI. The significant abundance coefficients from the differential test conducted with Corncob for Thyroid and Statin medication users are reported in S4 and S5 Tables, respectively. Of the above genera identified as differentially abundant by Corncob, only *Neisseria* showed a trend ($P$ = 0.057) in Statin+ users by MaAsLin2 and no genera were identified using both ALDEx2 and ANCOM-II (S5 Table).

## Discussion

Research over the past several years has demonstrated that commonly prescribed medications can alter the diversity/composition of the gut microbiota [4–10] as well as produce side effects and alter drug efficiency [1–3]. Much of the previous literature on gut microbes has focused specific single medications [4–8]. For example, two studies published in the journal GUT in 2016 reported the influence of PPI use on the gut microbiome [4, 5]. They both reported lower gut microbial alpha diversity and an increase in the *Streptococcaceae* family with PPI use. Interestingly, bacteria that are typically found in the oral cavity are increased in the gut microbiome of PPI users [5], suggesting a potential role for oral microbes to influence the health of

(A)

(B)

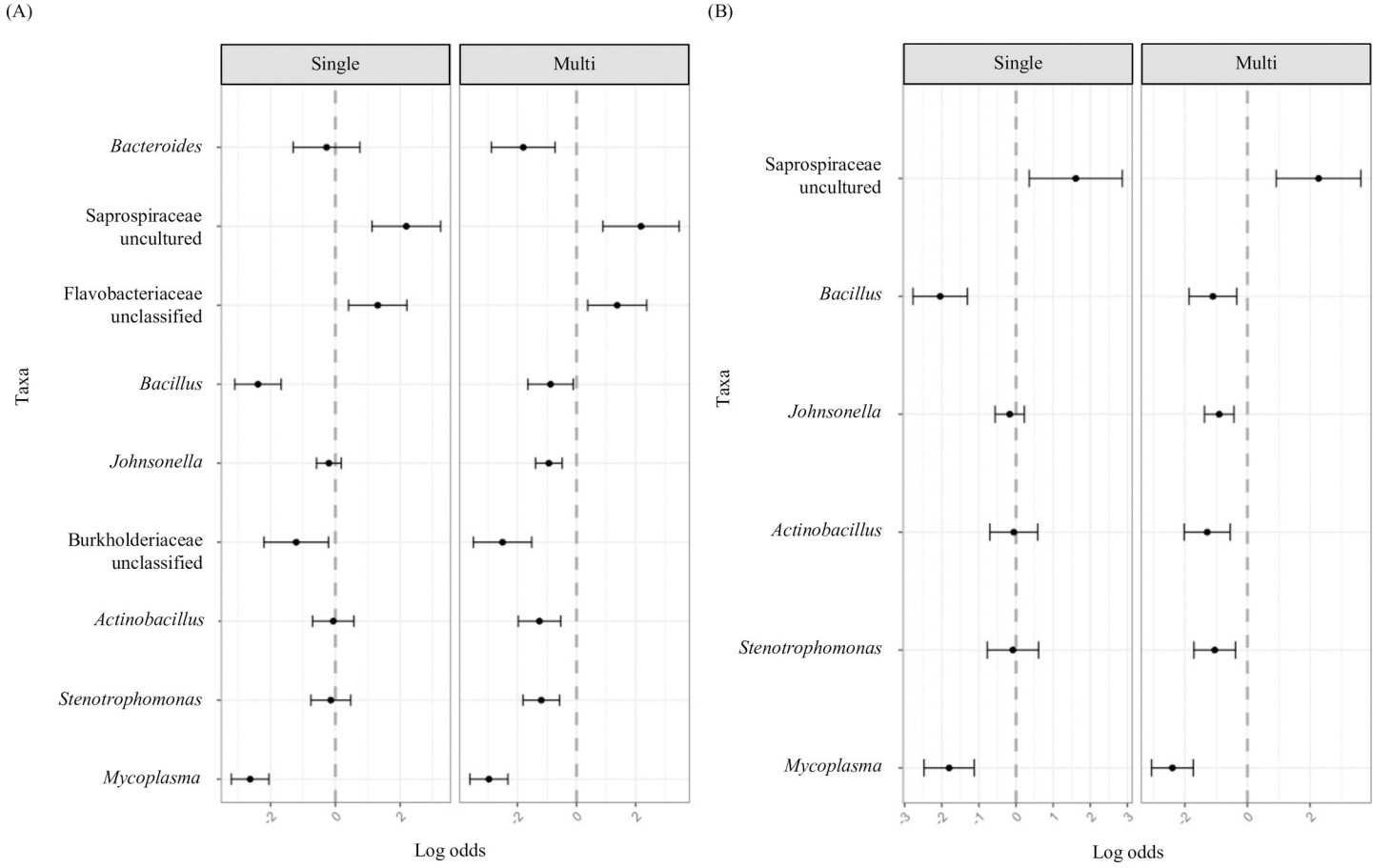

**Fig 3.** Differentially abundant genera in single and multi-medication users compared to non-medication users in (A) covariate unadjusted and (B) covariate adjusted models. Covariates include sex, age, and BMI. Results were adjusted by False Discovery Rate (FDR) using Benjamini and Hochberg method and genera meeting an FDR of q = 0.1 are presented. Non-medication users (n = 546); "Single" represents participants taking only one medication at ATC code Level 4 (n = 276); "Multi" represents participants taking 2 or more medications at ATC code Level 4 (n = 225).

non-oral areas of the body. Furthermore, gut microbes may not be the only human microbiome influenced by medication [11–13]. Recent research demonstrates that medications such as PPIs may also modify the oral microbiota [12, 24]. In addition, multiple medications are known to cause oral side effects, thus it is probable that other commonly used medications may influence the oral microbial community as well. Oral microbial communities are especially diverse in terms of community membership or species richness [39] and is predominated by *Streptococcus*, *Veillonella*, *Prevotella*, and *Neisseria* [21, 40]. Although the oral microbiome plays an important role in health and disease [14, 18–20], research on medication use and the oral microbiota is very limited.

While many studies have reported changes in the gut microbiota with commonly used medications [9, 10] much less is known about the microbes of the oral cavity in relation to commonly used medications. Therefore, a cross-sectional observational study was conducted to assess the associations between medication use and the oral microbiome in a Canadian cohort. Nearly half of the participants reported taking at least one prescription medication with the most common medications being thyroid hormone, statin, and PPI medications. Both alpha and beta diversity measures showed similar diversity patterns among single medication users, multi-medication users, and those taking no medication (Figs 2 and 3). But single

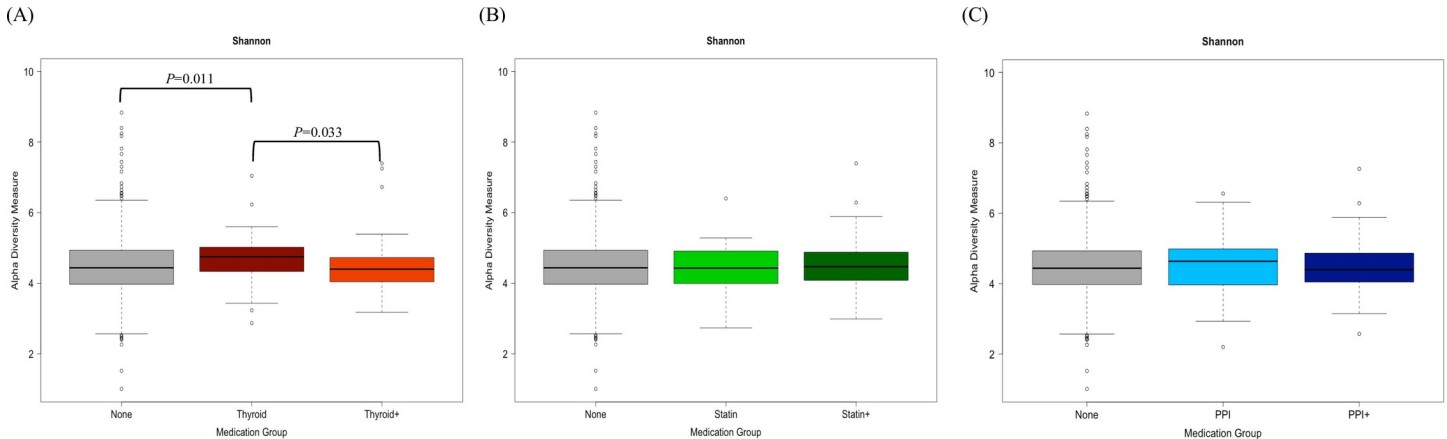

**Fig 4.** Shannon diversity index among (A) Thyroid Hormone, (B) Statin, (C) PPI users compared to participants taking no medication. Compared using a Kruskal-Wallis test and if necessary, followed with Dunn's test. P-values above box plots indicate the results of Dunn's tests with Bonferroni correction. None represents participants taking no medications (n = 546); Thyroid represents participants only taking Thyroid Hormone medication (n = 54); Thyroid+ represents participants taking Thyroid Hormone medication plus other medication(s) (n = 58); Statin represents participants only taking Statin medication (n = 30); Statin+ represents participants taking Statin medication plus other medication(s) (n = 65); PPI represents participants only taking PPI medication (n = 31); PPI+ represents participants taking PPI medication plus other medication(s) (n = 61).

and multi-medication users displayed several genera (Saprospiraceae uncultured, *Bacillus*, *Johnsonella*, *Actinobacillus*, *Stenotrophomonas*, and *Mycoplasma*) that were significantly enriched or depleted compared to non-medication users (Fig 3). Of the above noted genera, *Bacillus* was significantly depleted in both thyroid hormone and statin users (S3 Fig). When exploring specific medications in our initial analysis, we found the relative abundance of several genera to significantly differ in participants taking thyroid hormones or statins compared to participants taking no medications however, the majority of differences had small effect sizes. Our recent work demonstrates that differential abundance methods produce variable results [34] and since effect sizes in the current study were small, we analyzed our data using

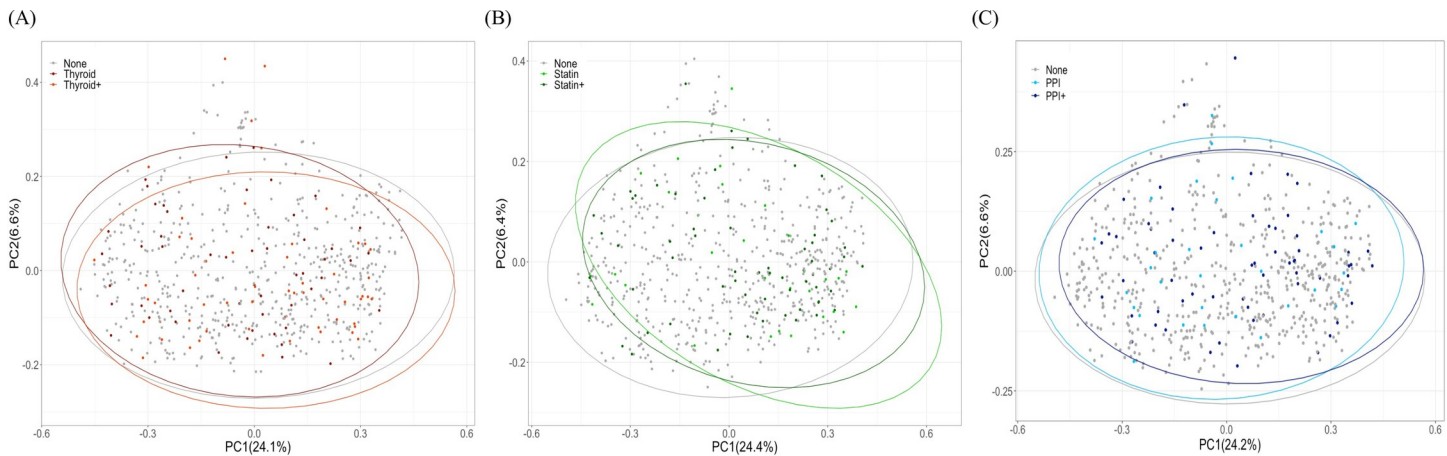

**Fig 5.** Beta diversity analyses among (A) thyroid, (B) statin, and (C) PPI users and non-medication users are represented by Principal Coordinates Analysis plots based on Bray-Curtis dissimilarity. None represents participants taking no medications (n = 546); Thyroid represents participants only taking Thyroid Hormone medication (n = 54); Thyroid+ represents participants taking Thyroid Hormone medication plus other medication(s) (n = 58); Statin represents participants only taking Statin medication (n = 30); Statin+ represents participants taking Statin medication plus other medication(s) (n = 65); PPI represents participants only taking PPI medication (n = 31); PPI+ represents participants taking PPI medication plus other medication(s)(n = 61).

three additional differential abundance tools. However, no taxa were recovered using the additional tools, indicating only minor evidence for shifts in the abundance of these taxa.

As mentioned above, much of the literature published on medication use and the microbiome has focused on single medication use [4–8]; however, there are several situations or disease conditions where an individual may be taking multiple medications. The use of polypharmacy is higher in particular groups of people such as cancer survivors or patients with specific chronic conditions like IBD [41–43]. Even in the current study which excluded individuals with major chronic conditions, over 40% of participants taking prescription medications reported taking more than 1 type of medication. A recent study by Vila and colleagues examined the relationship between the gut microbiome and commonly used medications [10]. In that study, the authors examined 3 cohorts (a general population, IBD, and IBS cohort) and found that the median number of medications was 0, 2, and 1 for each cohort, respectively (overall range = 0–12 medications per participant) but over 500 different combinations of medications were reported. There were no significant changes in microbial richness per number of medications used (alpha-diversity) but there were significant differences in beta diversity and the number of medications used in all cohorts [10]. In contrast to the findings by Villa et al., the current study of the oral microbiome found no differences in alpha or beta diversity between general medication use (single and multi-medication users; Figs 1 and 2).

In the current study, the use of thyroid hormones (thyroxine) resulted in a significant increase in the alpha diversity of the oral cavity, whereas previous reports of gut microbial diversity in patients taking the thyroid hormone thyroxine showed no statistical differences from controls (not receiving thyroxine) [44]. Khan and associates investigated the impact of statins on gut microbiome composition and reported differences in alpha and beta diversity measures, with more variability in the untreated hypercholesterolemic patients (n = 15) compared to statin-treated hypercholesterolemic patients (n = 27) or non-hypercholesterolemic individuals (n = 19) [45]. The study also showed that statin-treatment shifted alpha diversity in the gut closer to non-hypercholesterolemic individuals and was dominated by a higher relative abundance of the families Ruminococcaceae (Clostridia, Firmicutes) and Verrucomicrobiaceae (class, Verrucomicrobia) and the species *Faecalibacterium prausnitzii (clostridia, Firmicutes)* [45], two of which belong to the phyla Firmicutes and highly prevalent in the gut. In the current oral microbiome study, statin use (n = 111) was also associated with small shifts in oral microbial composition of specific genera that belong to Firmicutes phyla such as *Bacillus*, *Catonella, and Johnsonella* (S3B Fig), but failed to observe significant changes in alpha diversity of the oral microbiome (Figs 4B and 5B). Similarly, Villa at al. did not observe any significant changes in alpha with statin use but did report significant differences in beta diversity with specific medications including statins and PPIs [10].

Previously published research on the influence of PPIs on the gut microbiome has displayed mixed results on alpha diversity, reporting either lower alpha diversity or no difference [4, 5, 10]. In a small study of the oral microbiome (n = 10 participants), treatment with PPI (20mg esomeprazole) for 4 weeks was shown to significantly decrease the relative abundance of salivary *Neisseria* and *Veillonella* in healthy individuals (as defined by no medical treatments or probiotics within 3 months) [12]. The study also showed that both Shannon diversity was lower after PPI usage, and there were significant differences in beta diversity between PPI users and non-users as determined by weighted UniFrac and Bray-Curtis distance multivariate analysis [12]. The current study did not observe these same differences in alpha and beta diversity measures, but there are several differences between the studies worth noting, such as study design (intervention vs observational cross-sectional), sample and grouping size (n = 10 vs n = 1000+ and the number of participants in control vs medication groups), the definition of 'healthy', dose and duration (unknown in the current study, possibly years), as well as the type

of PPI. In the current study, all types of PPI were considered and the most commonly reported were omeprazole, esomeprazole, pantoprazole, and rabeprazole.

The presence of certain diseases or conditions is worth noting when examining the influence of medications on the oral microbiome. In 2021 Kawar and colleagues reported findings from a cross-sectional study examining the oral microbial communities of gastroesophageal reflux disease (GERD) patients with or without PPI use compared to healthy controls (defined as free of GERD and not using PPI) [24]. The study showed no significant differences in alpha diversity among the groups but did report taxa that were significantly different between the controls and GERD patients not taking PPI, and when GERD patients taking PPI were compared to controls there were no taxa that showed significant difference. The microbial profiles of GERD patients taking PPI look more similar to the health controls (no disease, no PPI use). It is unclear if these shifts in microbial populations observed with PPI use cause direct health benefits, but when studying specific diseases or conditions, it does stress the importance of considering if patients are currently receiving treatment or not.

Shifts in specific oral taxa have been noted in multiple non-oral host diseases such as diabetes, cancer, and atherosclerosis [18, 46], suggesting a relationship between oral dysbiosis and systemic disease. The management of many chronic conditions or clinical symptoms often requires the use of one or more prescription medications, which may perturb the oral microbiome. For example, Yang and colleagues compared the salivary microbiome of treatment-naïve diabetic patients to those treated with metformin alone or a combination treatment (insulin plus metformin or other hypoglycemic drugs) [47]. Compared to treatment-naïve patients, those taking metformin showed significant changes in 11 genera and the combination treatment showed significant changes in 15 genera and noticeably changes in only 3 genera (*Blautia*, *Cobetia* and *Nocardia*) were similar between the two treatment groups. Neither the alpha diversity nor the beta diversity of the salivary change significantly with metformin or combined treatment but diversity measures were significantly different between nondiabetic and diabetic patients. Significant differences were noted at the phylum, genus and species level of nondiabetic individuals compared to treatment-naïve diabetic patients [47]. This work highlights changes in salivary bacteria between individuals with and without diabetes but also differential effects of single or combination medication treatment.

The studies discussed above emphasize some important considerations for future research and stress some limitations and strengths of the current study. One challenge of our work and others in the field is the reliance on self-reported data. Self-reported data are limited by social desirability and recall bias, however self-report offers a noninvasive, minimal patient burden means to access large amounts of personal data in large populations. Self-reported medication data shows overall agreement with prescription databases, although variations among different drug classes is worth consideration [48–50]. Another consideration for self-reported medication data is the amount of information collected. For instance, this data often lacks greater level of detail on history, dosage, duration, and route of administration. In the current study, participants were only asked about current prescription medication use and requested to provide the name of the medication and the DIN however, most participants only provided the general drug name, making it difficult to code past level 4 of the ATC or dosage and route of administration. On the other hand, the use of self-reported questionnaire data often includes the collection of a large about of demographic and lifestyle data along with health information. Participants of the Atlantic PATH cohort examined in this study completed questionnaire data on various demographic, anthropometric, and lifestyle factors including smoking, alcohol use, physical activity, and diet. Smoking has previously been shown to alter alpha and beta diversity as well as several taxa of the oral microbiome [51, 52] and thus was used as exclusion criteria for the current study. Our previous study on the healthy oral microbiome explored

demographic, anthropometric, and lifestyle factors (including diet, alcohol use, and physical activity) and showed that several anthropometric measurements as well as age and sex, were associated with overall oral microbiome structure but individually each factor was associated with only minor shifts in the overall taxonomic composition of the oral microbiome [21]. With this knowledge, confounding factors such as sex, age, and BMI were controlled for during analysis in the current analysis. Finally, underlying health conditions should be considered because the composition and the diversity of both the gut and oral microbiomes differs in many chronic diseases [9, 10, 14, 18–20]. Participants in the current study self-reported major chronic conditions by answering a set of closed ended questions on several different diseases, but the list was not comprehensive. Importantly these participants will be followed for up to 30 years by linkage to administrative health databases in the future. Moving forward this will provide a wealth of information related to disease including a broader range of conditions, disease activity and therapeutic approaches. Since microbes can metabolize a wide range of different medications [53] and have the potential to alter their mechanism of action, future research on the interaction between medications and the microbiome in specific diseases is necessary to provide insight into anticipated therapeutic outcomes.

## Conclusions

In conclusion, our study shows that at the genus level the oral microbiome is relatively similar between individuals with no major chronic conditions taking commonly prescribed medications with some evidence indicating shifts in a minor number of taxa. Data from this cross-sectional study may be useful for designing future studies with more focused questions on specific types of medications, dosage, duration, and route of administration. The oral microbiome offers a potential tool for screening health status, forecasting future disease risk or predicting treatment outcomes. However, we need to gain a better understanding of the effects of specific diseases on the oral microbiome as well as the influence of commonly prescribed medications for those diseases. Future longitudinal studies with linkage to provincial Drug Information Systems are necessary to study patients before and after administration of specific medications along with appropriate controls in large cohorts.

## Supporting information

**S1 Fig. Microbial diversity among participants only taking thyroid hormones, statins, or PPIs compared to participants taking no medication.** Alpha diversity represented by (A) Shannon diversity and (B) Evenness, and beta diversity as represented by (C) Bray-Curtis dissimilarity, and (D) unweighted UniFrac.
(PDF)

**S2 Fig.** Beta-diversity represented by Principal Coordinates Analysis plots based on weighted uniFrac distances among participants taking no medication and those taking (A) thyroid, (B) statin, and (C) PPI.
(PDF)

**S3 Fig.** Differentially abundant genera in (A) thyroid hormone users, and (B) statin users compared to participants taking no medication, controlling for covariates (sex, age, and BMI).
(PDF)

**S1 Table. Characteristics of participants taking none, one, or multi medications.**
(PDF)

**S2 Table. Characteristics of participants taking the most frequently reported medications alone or in combination.**
(PDF)

**S3 Table. Differentially abundant genera in saliva of single and multi-medication users.**
(PDF)

**S4 Table. Differentially abundant genera in saliva of thyroid medication users.**
(PDF)

**S5 Table. Differentially abundant genera in saliva of statin medication users.**
(PDF)

## Acknowledgments

Thank you to all participants of the Atlantic PATH project for donating their time, personal health information and biological samples to this project, and to Atlantic PATH team members for data collection and management. The data used in this research were made available by the Atlantic Partnership for Tomorrow's Health (Atlantic PATH) study, which is the Atlantic Canada regional component of the Canadian Partnership for Tomorrow Project funded by the Canadian Partnership Against Cancer and Health Canada. The views expressed herein represent the views of the authors and do not necessarily represent the views of Health Canada. JTN is funded by both a Nova Scotia Graduate Scholarship and a ResearchNS Scotia Scholars award.

## Author Contributions

**Conceptualization:** Vanessa DeClercq, Jacob T. Nearing, Morgan G. I. Langille.

**Data curation:** Vanessa DeClercq, Jacob T. Nearing.

**Formal analysis:** Vanessa DeClercq, Jacob T. Nearing.

**Methodology:** Vanessa DeClercq.

**Resources:** Jacob T. Nearing.

**Supervision:** Morgan G. I. Langille.

**Visualization:** Vanessa DeClercq.

**Writing – original draft:** Vanessa DeClercq.

**Writing – review & editing:** Vanessa DeClercq, Jacob T. Nearing, Morgan G. I. Langille.

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
