## [Decision Letter · Decision Letter 0]

13 Sep 2021

PONE-D-21-20779Commonly used medications have little impact on the oral microbiome of individuals living without major chronic conditionsPLOS ONE

Dear Dr. DeClercq,

Thank you for submitting your manuscript to PLOS ONE. After careful consideration, we feel that it has merit but does not fully meet PLOS ONE’s publication criteria as it currently stands. Therefore, we invite you to submit a revised version of the manuscript that addresses the points raised during the review process.

We look forward to receiving your revised manuscript.

Kind regards,

Christopher Staley, Ph.D.

Academic Editor

PLOS ONE

Journal Requirements:

2. Please include additional information regarding the survey or questionnaire used in the study and ensure that you have provided sufficient details that others could replicate the analyses. For instance, if you developed a questionnaire as part of this study and it is not under a copyright more restrictive than CC-BY, please include a copy, in both the original language and English, as Supporting Information

Additional Editor Comments (if provided):

Clarification of the rationale and addition details regarding the patient demographics/medication history are needed.

Reviewers' comments:

Reviewer's Responses to Questions

**Comments to the Author**

1. Is the manuscript technically sound, and do the data support the conclusions?

Reviewer #1: Yes

Reviewer #2: Partly

2. Has the statistical analysis been performed appropriately and rigorously? 

Reviewer #1: Yes

Reviewer #2: Yes

3. Have the authors made all data underlying the findings in their manuscript fully available?

Reviewer #1: Yes

Reviewer #2: No

4. Is the manuscript presented in an intelligible fashion and written in standard English?

Reviewer #1: Yes

Reviewer #2: Yes

5. Review Comments to the Author

Reviewer #1: This manuscript examines the impact of common medications on the oral microbiome. The author’s premise is based on published data that illustrates the impact of several medications, including PPI on both the gut and oral microbiomes. The study design was cross-sectional, with over 8,000 saliva samples from a Canadian national prospective study that examined the influence of genetic, environmental, and lifestyle factors in the development of chronic disease. The study was well-designed, with acceptable rigor in generation and multivariate analysis of the microbiome data and metadate from each subject. Overall, the study did not identify significant changes in oral microbiome diversity or composition due to subject use of single or multiple medications.

Comments:

1) The manuscript should include the authors’ rationale for using the cross-sectional study design. As they note, other studies which examine changes in the oral microbiome before and after medication have identified differences in microbiota composition and diversity. Notably, this difference occurs with PPI, a medication the authors examine in this current study.

2) Additional details are needed regarding the analysis of statin and PPI as the single medication. Where these chosen after statistical evaluation with all single and combination drugs examined?

3) The authors appear to include all medications listed in Table 1 in their analysis of the multi-medication users. Did they consider the possibility of interactions between these drug combinations that may affect their analysis?

4) Were any of the medications listed in Table 1 administered directly into the oral cavity? For example, steroid-based inhalers for asthma, with the steroid in direct contact with the oral mucosal surfaces.

Reviewer #2: Title, abstract and introduction:

1. This title could be improved. It is not an accurate conclusion of the study findings. The “little impact” was not accurate since there were some significant altered relative abundances in some taxa across groups; also, there are some limitations on the study design and grouping. I suggest a title like “investigation of impact of commonly used medications on the oral microbiome of individuals without major chronic conditions”.

2. Add a reference for line 58-60

3. What was the author’s hypotheses for the current study? Was there a change expected to be observed across groups?

Method:

1. The study groups were defined as no-current medication use (None); currently use 1 medication(single); currently use more than 1 medication(multi), but was the duration of use was examined? Duration of use could also be confounder of the analysis results.

2. How were the current heath stage of the participants regulated? They may not have major chronic diseases but possibly have other conditions.

3. I’m not sure I understand the definition of “most commonly used medication”. Is that possible that subject in the Single group were taking all different medication? Does that mean that those medication were all from one category? How the variation between different medications was controlled? Please clarify in the “Medication Data” section.

4. Line155-156: It was mentioned that 1214 saliva samples were collected. How many samples left after filtering based on total reads generated? Please clarify.

5. From what I understand, diversity analysis was done using the rarefied data with some sample filtered out because not met the sample depth cutoff, and differential abundance testing was based on non-rarefied data. Was the sample size equal for those analysis? Please make it clear.

Results:

1. In the single user group and multi-medication user group, how many medication classes are there in each, how many subjects in each medication class in each group? Please clarify. A table shown sample size per medication class in across study groups would be useful.

2. Were any pair-wise tests for groups done for alpha diversity and beta diversity? any significance?

3. Did the authors compare between different medication classes? For example, Thyroid Hormone vs Stain vs PPI?

4. The plots could be colored differently by groups and by comparisons also? For example, figure4; 5; and S1, same colors have been used for a different comparison (non vs single vs multi).

Discussion and conclusion:

The first paragraph was just repeating of the results. I’d recommend this order for the discussion: previously reported importance of commonly used medication on gut microbiome and why oral microbiome is also worth to be studied (link to gut microbiome); summary of current findings –although no major impact but still need to focus on the significant altered taxa identified; then talk about results from other similar studies; and focus on the limitation of the current study and how the results might be confounded.

Study limitations such as duration of drug use and health status (you may also want to consider oral/periodontal health; participants’ behavior regulation– smoking/drinking) need to be highlighted in the discussion and conclusion.

6. PLOS authors have the option to publish the peer review history of their article (what does this mean?). If published, this will include your full peer review and any attached files.

Reviewer #1: No

Reviewer #2: No

---

## [Author Response · Author response to Decision Letter 0]

27 Oct 2021

Additional Editor Comments:

Clarification of the rationale and addition details regarding the patient demographics/medication history are needed.

Response: 

Rationale – The oral microbiome plays a role in human health, both in maintaining oral health homeostasis and contributing to local and systemic conditions. Multiple medications have been reported to impact oral health causing symptoms such as dry mouth, lesions, ulcers, and altered taste, thus it may be possible that the medication are disrupting the oral microbial community. Previous research has demonstrated that commonly used medications alter the gut microbiota but research on commonly used medications and the oral microbiome is extremely limited. Since the oral microbiome plays an important role in health and disease, alteration of the microbiota by commonly used medications could have unintended consequences on human health. Thus, we aimed to investigate the role of commonly used medications on the composition and diversity of the oral microbiota (p.6).

Additional participant demographics have been previously published [Nearing et al. 2020] (p.7). While we agree that medication history could improve our analysis unfortunately, the Atlantic PATH cohort did not collect this information. Only current medication use was captured in the questionnaires. Participants were asked to provide the name of the medication they were taking as well as provide the drug identification number. This information along with further coding of the medication can be found on p.8.

Reviewer Comments to the Author

Reviewer #1: 

1) The manuscript should include the authors’ rationale for using the cross-sectional study design. As they note, other studies which examine changes in the oral microbiome before and after medication have identified differences in microbiota composition and diversity. Notably, this difference occurs with PPI, a medication the authors examine in this current study.

Response: 

Previous work on the influence of commonly used medications on the oral microbiota is extremely limited. As a starting point, the use of a cross-sectional study design allowed us to get a quick snapshot of the oral microbiome from a population-based cohort while comparing multiple drugs at the same time and controlling for other variables such as sex, age, and BMI. While this type of study cannot infer cause and effect, it allows us to scan a large cohort to identify associations between medication classes and oral microbiota composition. This type of data is useful for designing future longitudinal studies on specific types of medications where we want to answer much more specific questions regarding medication dose, duration, route of administration, etc. Medication data for this study was accessed from the Atlantic PATH cohort for which they only have basic medication use at baseline. However, the future linkage to provincial Drug Information System will allow us to ask much more detailed questions in the future and assess changes overtime. We have included a rationale for the study design (p.6) and how this type of data is useful for development of future studies, including longitudinal (p.26 &27).

2) Additional details are needed regarding the analysis of statin and PPI as the single medication. Where these chosen after statistical evaluation with all single and combination drugs examined?

Response: 

All medications that were reported by participants were first coded according to the Anatomical Therapeutic Chemical (ATC) Classification System. Participants were then grouped as none, single, or multi medication users according to classification at the fourth level of the ATC code. Participants that were taking one unique medication at the 4th level of the ATC code were classified as a ‘single’ medication user. Participants taking 2 or more unique medications at the 4th level of the ATC code were classified as a ‘multi’ medication users. Participants that completed the questionnaire without listing any medications were assumed to not be taking any medications and classified as ‘none’. Subsequently, the frequency of each reported medication at the 4th level of the ATC code was assessed to determine the mostly commonly reported medications. Medications that were reported more than 5 times by participants are listed in Table 1, with thyroid hormone medications, proton pump inhibitors, and HMG CoA reductase inhibitors being the 3 most frequently reported. Those that were taking the most frequently reported medications were further divided into those that were only taking the specific medication or those that were taking the specific medication plus another medication (eg. Thyroid, Thyroid+, Statin, Statin+, PPI and PPI+). These groups and the general none, single, and multi-medication groups were used for further statistical on oral microbial composition and diversity. This above information has been incorporated into the methods section ‘Medication data’ (p.8).

3) The authors appear to include all medications listed in Table 1 in their analysis of the multi-medication users. Did they consider the possibility of interactions between these drug combinations that may affect their analysis?

Response: 

Yes, this is an important consideration, and it is likely that interactions between medications could influence microbial composition. Because of this we have included both participants that have indicated they are taking one single medication as well as those taking multiple medications. Ideally, we would look at specific medication interactions (not just lump into ‘multi’ or other) however, some participants were taking up to 15 medications resulting in numerous different combinations of medications. The analysis becomes complex and unfortunately groups become too small to run statistical analysis. But the possible interaction between medications in intriguing and to partially address this issue we assessed single medication users (e.g. Statins) and/or those that were taking a specific medication plus another medication (e.g. Statin+). We acknowledge that these medication groups were not described well in the original version of the manuscript and additional detail has been added to the methods section (p.8). 

4) Were any of the medications listed in Table 1 administered directly into the oral cavity? For example, steroid-based inhalers for asthma, with the steroid in direct contact with the oral mucosal surfaces.

Response: 

Good question. Yes, many of the medications were taken orally and would have come in direct contact with the oral mucosal surfaces however, some medications can be taken via multiple routes. For example, proton pump inhibitors can be taken orally or intravenously, whereas statins are only taken orally. Unfortunately, the level of detail captured on the questionnaires by participants self-reporting medication use is limited and most participants only provided a general drug name, making coding past level 4 difficult. More detailed level of information from linkage to provincial drug information systems would be necessary to allow for the analysis at more detailed level, unfortunately that information was not available. Using the level 4 ATC code, we have added the possible routes of administration for the class of medications (Table 1) and included a note in the limitations (p.25-26) section as well as in the conclusions section (p.27) about the need for future studies to consider route of administration and a greater level of by linking to health system databases. 

Reviewer #2: 

Title, abstract and introduction:

1. This title could be improved. It is not an accurate conclusion of the study findings. The “little impact” was not accurate since there were some significant altered relative abundances in some taxa across groups; also, there are some limitations on the study design and grouping. I suggest a title like “investigation of impact of commonly used medications on the oral microbiome of individuals without major chronic conditions”.

Response: 

We have revised the title to the above suggested.

2. Add a reference for line 58-60

Response: 

Added (p.4, line 59).

3. What was the author’s hypotheses for the current study? Was there a change expected to be observed across groups?

Response: 

We hypothesized that the use of medications would be associated with changes in oral microbiota composition and diversity, and further augmented with the use of multiple medications. We have added our hypothesis at the end of the introduction section (p.6)

Method:

1. The study groups were defined as no-current medication use (None); currently use 1 medication(single); currently use more than 1 medication(multi), but was the duration of use was examined? Duration of use could also be confounder of the analysis results.

Response: 

We agree, this would have been a very valuable piece of information. Unfortunately, this information was not collected by the cohort, only the name of the medication and drug identification number (DIN) were provided. We have included more detail about the medication information that was provided and the coding (p. 8) as well as limitations of the current study (p.25-26) and the need for future studies to examine medication duration (p.27).

2. How were the current heath stage of the participants regulated? They may not have major chronic diseases but possibly have other conditions.

Response: 

The questionnaire completed by participants of the Atlantic PATH cohort included a section on personal medical history. Participants were asked to respond (yes, no, don’t’ know) to questions about various health conditions (all conditions are listed on p.7). It is possible that participants could have conditions other than those included in the questionnaire. Unfortunately, we only had access to responses from the questionnaires for this study. The Atlantic PATH cohort is in the process of linking to administrative health databases however, the data is not yet available. This is another limitation and an important consideration for future research (p.25-26). 

3. I’m not sure I understand the definition of “most commonly used medication”. Is that possible that subject in the Single group were taking all different medication? Does that mean that those medication were all from one category? How the variation between different medications was controlled? Please clarify in the “Medication Data” section.

Response: 

The most commonly used medication is actually the most frequently reported medication. Those in the Single group could be taking a wide range of medications, ‘Single’ means the participant reported taking only one medication at the ATC code level 4 (it can be any medication, but only one). When we examine specific medications, we focused on medications from one category such as statins. But we know that not all the statin users were only taking one medication and we were concerned about interactions between medications. Therefore, participants that were taking the most frequently reported medications were further divided into those that were only taking the specific medication or those that were taking the specific medication plus any other medications (eg. Thyroid, Thyroid+, Statin, Statin+, PPI and PPI+). We acknowledge that the Medication Data section was lacking detail in the original submission. We have now revised that section to include more details on medication groups, coding, and how the most commonly used medications were identified (p.8). 

4. Line155-156: It was mentioned that 1214 saliva samples were collected. How many samples left after filtering based on total reads generated? Please clarify.

Response: 

16S samples that had a sequencing depth of less than 5000 reads were removed from the analysis. Thus, the final number of samples for analysis was 1,049. We have included this information in the “16S rRNA gene sequencing” and ‘Statistical Analysis” sections (p.9) as well as provided sample size for each analysis in the figure legends.

 5. From what I understand, diversity analysis was done using the rarefied data with some sample filtered out because not met the sample depth cutoff, and differential abundance testing was based on non-rarefied data. Was the sample size equal for those analysis? Please make it clear.

Response: 

We have added additional detail to the “16S rRNA gene sequencing” and ‘Statistical Analysis” sections (p.9-11) as well as provided sample size for each analysis in the figure legends.

Results:

1. In the single user group and multi-medication user group, how many medication classes are there in each, how many subjects in each medication class in each group? Please clarify. A table shown sample size per medication class in across study groups would be useful.

Response: 

At ATC code level 4 there were 144 medication classes reported over, 72 medication classes in the single medication group and 129 in the multi medication group. We have revised Table 1 to include overall counts per medication class as well as counts for single and multi-medication users (Table 1). 

2. Were any pair-wise tests for groups done for alpha diversity and beta diversity? any significance?

Response: 

Yes, when the overall test for alpha or beta diversity was statistically different, pair-wise tests were used to determine which comparisons differed statistically. For example, the Dunn’s test is a pairwise comparison that was used when we rejected the Kruskal-Wallis such as in Figure 4a. When the Dunn’s test with Bonferroni correction was performed for alpha diversity, the P-values for the pairwise comparison are shown above the brackets in the figure. For beta-diversity, when the results of the PERMANOVA were statistically significant, pairwise comparisons with Bonferroni correction were applied. We have edited the statistical analysis section to clarify the use of pair-wise testing (p.9-11) and the results of pairwise comparisons can be found on p.17 & 18. 

3. Did the authors compare between different medication classes? For example, Thyroid Hormone vs Stain vs PPI?

Response: 

We have included a comparison of microbial diversity between participants taking only thyroid, statins, PPI, or none (S1Fig). Some measures of alpha diversity differed among thyroid hormone users (Shannan diversity P=0.041; Evenness P=0.013; S1a and S1b Fig) but beta diversity was non-significant among participants taking only thyroid hormones, statins, PPIs or no medications (Bray-Curtis P=0.104; weighted UniFrac p=0.324; S1c and S1d Fig). This information can now be found on p. 17.

4. The plots could be colored differently by groups and by comparisons also? For example, figure4; 5; and S1, same colors have been used for a different comparison (non vs single vs multi).

Response: 

We have edited the plots so that groups are colored differently and consistent throughout the manuscript. Participants taking no medications are always shown in dark grey. General single medication users are shown in royal blue and multi medication users in golden-orange (Fig 2 &3). Thyroid and Thyroid+ users are shown in dark red and dark orange; Statin and Statin+ users in light green and dark green; PPI and PPI+ users in light blue and dark blue (S1 Fig, S2 Fig, Fig 4, Fig 5).

Discussion and conclusion:

The first paragraph was just repeating of the results. I’d recommend this order for the discussion: previously reported importance of commonly used medication on gut microbiome and why oral microbiome is also worth to be studied (link to gut microbiome); summary of current findings –although no major impact but still need to focus on the significant altered taxa identified; then talk about results from other similar studies; and focus on the limitation of the current study and how the results might be confounded.

Response: 

The discussion section has been reorganized, starting with a new paragraph on the importance of medication use on the gut microbiome and link to oral microbiome, followed by a summary of current findings and then a discussion about similar studies (starting with general medication use, then going through specific medications), and finally ending with a limitations section (p.20-27).

Study limitations such as duration of drug use and health status (you may also want to consider oral/periodontal health; participants’ behavior regulation– smoking/drinking) need to be highlighted in the discussion and conclusion.

Response: 

We had previously pointed out some limitations throughout the study but have now refocused the discussion so that the limitations are discussed together in the last paragraph before the conclusions section (p.25-27).

---

## [Decision Letter · Decision Letter 1]

23 Nov 2021

Investigation of the impact of commonly used medications on the oral microbiome of individuals living without major chronic conditions

PONE-D-21-20779R1

Dear Dr. DeClercq,

We’re pleased to inform you that your manuscript has been judged scientifically suitable for publication and will be formally accepted for publication once it meets all outstanding technical requirements.

Kind regards,

Jonathan Jacobs

Academic Editor

PLOS ONE

Additional Editor Comments (optional):

Reviewers' comments:

Reviewer's Responses to Questions

**Comments to the Author**

1. If the authors have adequately addressed your comments raised in a previous round of review and you feel that this manuscript is now acceptable for publication, you may indicate that here to bypass the “Comments to the Author” section, enter your conflict of interest statement in the “Confidential to Editor” section, and submit your "Accept" recommendation.

Reviewer #1: All comments have been addressed

Reviewer #2: All comments have been addressed

2. Is the manuscript technically sound, and do the data support the conclusions?

Reviewer #1: Yes

Reviewer #2: Yes

3. Has the statistical analysis been performed appropriately and rigorously? 

Reviewer #1: Yes

Reviewer #2: Yes

4. Have the authors made all data underlying the findings in their manuscript fully available?

Reviewer #1: Yes

Reviewer #2: Yes

5. Is the manuscript presented in an intelligible fashion and written in standard English?

Reviewer #1: Yes

Reviewer #2: Yes

6. Review Comments to the Author

Reviewer #1: (No Response)

Reviewer #2: (No Response)

7. PLOS authors have the option to publish the peer review history of their article (what does this mean?). If published, this will include your full peer review and any attached files.

Reviewer #1: No

Reviewer #2: No

---

## [Editor Report · Acceptance letter]

1 Dec 2021

PONE-D-21-20779R1 

Investigation of the impact of commonly used medications on the oral microbiome of individuals living without major chronic conditions 

Dear Dr. DeClercq:

I'm pleased to inform you that your manuscript has been deemed suitable for publication in PLOS ONE. Congratulations! Your manuscript is now with our production department. 

Kind regards, 

on behalf of

Dr. Jonathan Jacobs 

Academic Editor

PLOS ONE